# Strain Virtual Sensing for Structural Health Monitoring under Variable Loads

**DOI:** 10.3390/s23104706

**Published:** 2023-05-12

**Authors:** Bartomeu Mora, Jon Basurko, Iman Sabahi, Urko Leturiondo, Joseba Albizuri

**Affiliations:** 1Ikerlan Technology Research Centre, Basque Research and Technology Alliance (BRTA), 20500 Arrasate-Mondragon, Spain; jbasurko@ikerlan.es (J.B.); uleturiondo@ikerlan.es (U.L.); 2Faculty of Engineering in Bilbao, University of the Basque Country (UPV/EHU), 48013 Bilbao, Spain; joseba.albizuri@ehu.eus; 3KU Leuven, Department of Mechanical Engineering, B-3001 Leuven, Belgium; iman.sabahi@kuleuven.be

**Keywords:** structural health monitoring, virtual sensing, Kalman filter, augmented Kalman filter, least squares estimation, strain virtual sensor

## Abstract

Virtual sensing is the process of using available data from real sensors in combination with a model of the system to obtain estimated data from unmeasured points. In this article, different strain virtual sensing algorithms are tested using real sensor data, under unmeasured different forces applied in different directions. Stochastic algorithms (Kalman filter and augmented Kalman filter) and deterministic algorithms (least-squares strain estimation) are tested with different input sensor configurations. A wind turbine prototype is used to apply the virtual sensing algorithms and evaluate the obtained estimations. An inertial shaker is installed on the top of the prototype, with a rotational base, to generate different external forces in different directions. The results obtained in the performed tests are analyzed to determine the most efficient sensor configurations capable of obtaining accurate estimates. Results show that it is possible to obtain accurate strain estimations at unmeasured points of a structure under an unknown loading condition, using measured strain data from a set of points and a sufficiently accurate FE model as input and applying the augmented Kalman filter or the least-squares strain estimation in combination with modal truncation and expansion techniques.

## 1. Introduction

Structure health monitoring (SHM) involves monitoring structures to determine their current condition. The use of SHM systems increases the safety of structural facilities and allows the optimization of the maintenance actions, predicting the remaining useful life of critical components and detecting anomalies that may indicate the presence of damage [1]. SHM systems require measured data from the structure using sensors, but it is not always possible to install all the necessary sensors at all the points of interest, either for technical or economic reasons. Virtual sensing (VS) allows obtaining measures from a system, not directly from physical sensors, but using data inference from other sensors [2]. The use of vs. in SHM systems results of interest takes place when it is necessary to obtain measurement data at points where it is not technically feasible to locate a real sensor, or when it is necessary to obtain measurements at a large number of locations, requiring a sensor network that is too extensive [3]. In conclusion, the use of vs. offers technical and economic advantages.

VS techniques can be classified into two main groups: data-driven techniques and model-based techniques [4]. Model-based techniques require a physics-based model capable of replicating the behavior of the monitored system. The model-based methods can be further classified into two groups: stochastic, where the system uncertainties are considered; and deterministic, where the system uncertainties are not considered [5,6].

Neural networks (NN) are commonly used in the data-driven vs. approaches. The NN are artificial intelligence algorithms that consist of complex networks of nodes (neurons) adjusted using training data, which relates the provided input data with the desired outputs. Artificial neural networks (ANNs) [7] and Convolutional neural networks (CNNs) [8] have been used for vs. applications. ANNs are simpler, because inputs are processed only in the forward direction, while CNNs are more complex because they use multiple types of layers for processing the provided input data.

Stochastic vs. algorithms have been commonly used for the estimation of states. One of the most known stochastic estimation algorithms is the Kalman filter (KF), a physics model-based algorithm proposed by R. Kalman in 1960 [9]. The KF uses a state-space model of the system to make state predictions (mean and covariance) based on information from the previous states. The KF is then a Bayesian estimator [10]. Input data from real sensors are used to correct the predictions and to update the algorithm parameters. Some examples of the use of the KF for strain estimations are found in [11,12]. An implementation of the KF with an augmented state-space model (which estimates the inputs together with the states of the system) was first proposed in 1969 by B. Friedland, to perform a state estimation with unknown inputs [13]. In 2010, E. Lourens et al. used the KF with the augmented state-space for dynamic force identification, and the following year they consolidated the so-called augmented Kalman filter (AKF), which has been used in later publications [14]. Other variants of the KF for nonlinear systems have been proposed: for example, the Extended Kalman filter (EKF) [15], that performs a linearization of the estimated mean and covariance for each time step, or the Unscented Kalman filter (UKF) [16], that avoids linearization by applying an unscented transformation to the estimated mean and covariance. An alternative to the Kalman filter is the Particle filter (PF) [17], which are also stochastic Bayesian estimators. For each time step, the PF generates multiple random estimations (particles) using Monte Carlo simulations. A weight is assigned to each particle, and the closest particles to the observation measurements are more weighted for the following time-steps.

Deterministic vs. algorithms have been used for force and strain estimation [3,18,19]. In this article, the least-squares state estimation (LSSE) is used. This method uses a Moore-Penrose pseudoinverse (a generalization of the matrix inverse which allows to obtain the pseudo-inverse matrix of a non-squared matrix [20]) to obtain the least-squares solution of the unknown strains [21]. Unlike the probabilistic methods, such as the previously described Kalman filters, the LSSE does not use the information of previous states and does not update its internal parameters to improve the estimation.

In this article, both stochastic and deterministic model-based algorithms are tested. The classic KF and the AKF are used as examples of stochastic methods and the LSSE is used as example of deterministic method. The AKF has been chosen because it is specifically designed to work without information of the external forces (which is of great interest for the work developed in this article), while the KF has been chosen to compare it with the AKF under the unknown external forces condition. On the other hand, the LSSE has been chosen due to its simplicity. EKF and UKF have been discarded because it is not intended to work with nonlinear models; meanwhile, the PF has been discarded due to its much higher computational complexity. NN algorithms are not used in this work because it is intended to avoid providing substantial amounts of training data. 

In recent years, Kalman filter and variants [22,23] and deterministic algorithms [24,25] have been used in SHM systems applied to structural facilities, such as wind turbines or bridges. In [22], the AKF is applied in a wind turbine to estimate the state of the structure and the external wind forces, using the turbine speed and the generator torque, together with accelerometer data, as input sensors. In [23], the KF algorithm is used for damage detection in aircraft frames and bridges, using accelerometers as input sensors. In [24], modal expansion is used for stress and strain estimation in an offshore structure prototype, using accelerometers and strain gauges as input sensors. In [25], modal expansion is used for strain estimation in a monopile offshore wind turbine, using accelerometers and strain gauges as input sensors too.

In order to work with strain measurements using the mentioned algorithms, as well as to obtain strain estimates from them, it is necessary to use the modal expansion/reduction method [18,26]. This method allows a number of strain measurements to be related to displacements in a model, and vice versa. Model reduction methods are also used to obtain lighter models from complex FE models, capable of being used by the vs. algorithms. Strain estimation is of interest due to its relationship with fatigue: by estimating the strain at critical points, the remaining useful life of a structure due to the accumulated fatigue can be estimated.

The main contribution of this article is to test different vs. algorithms (stochastic and deterministic) using real data obtained from a wide variety of experimental tests, obtained from an offshore wind turbine scaled prototype. For each selected algorithm, different input sensor configurations have been tested under different types of external forces applied in different directions (using an electromagnetic shaker installed on a rotating base on the top of the prototype), simulating variable loads on the prototype. The vs. algorithms are tested without measuring the applied forces, increasing the difficulty of the study.

This article is organized as follows: in Section 2, the modeling processes, the virtual sensing algorithms used, and the use case are described. In Section 3, the obtained experimental results are shown. In Section 4, the results are discussed and in Section 5, the final conclusions are presented.

## 2. Materials and Methods

### 2.1. System Modeling

In this subsection, the theoretical bases used on system modeling are described: mass-damper-spring equation, state-space formulation, and model discretization. The selected model reduction method, modal truncation, is also described.

#### 2.1.1. Finite Element Model

A Finite Element (FE) linear model of the monitored structure is created. Geometry, construction details and boundary conditions must be taken into account during the model creation. Mathematically, a FE model is defined by the mass-damper-spring second order differential Equation (MCK) Equation (1), that is able to describe the dynamical behavior of the model over time.
(1)$$M\ddot{q}\left(t\right)+C_{D}\dot{q}\left(t\right)+Kq\left(t\right)=f(t)$$

With n being the number of degrees of freedom (DoFs) of the model, **q**(t) is the displacement vector (with *n* × 1 dimension), **M**, **C_D_** and **K** are the stiffness, damping and mass matrices, respectively (with *n* × *n* dimension), and **f**(t) is the external forces vector (with *n* × 1 dimension).

The FE models of complex structures contain a large number of degrees of freedom (DoF), which implies that a big processing capacity and large amounts of time are needed to work with them. To remedy this issue, reduction methods need to be applied.

#### 2.1.2. Model Reduction

By applying model reduction methods to a full FE model, it is possible to obtain models with a much smaller number of DoFs, which are much lighter in terms of computation. It is a necessity when it is intended to work with FE models that represent complex structures (usually made of thousands or even millions of DoFs) and it is required to perform a high number of calculations over time (for example, a transient simulation) [27]. The reduced models can reproduce the dynamic behavior of the structure in limited ranges of use. Several model reduction methods can be found in the bibliography. Some examples are the Guyan static condensation [28], the improved reduced system (IRS) [29], the Craig-Bampton component mode synthesis [30] and the modal truncation [31]. In this article, the modal truncation is selected as model reduction method because it is a method that allows to maintain a great precision from the full model, within a defined range of use, and due to its simplicity of application [32].

To introduce the modal truncation method, first it must be explained that a dynamic model can be described through its mode shapes, using the mode-shapes matrix (**Φ**). Each column of **Φ** corresponds to an eigenvector (**φ**_i_), associated to an eigenvalue (λ_i_). The square root of every eigenvalue corresponds to a natural frequency of the system (ω_i_). The Φ-transformation implies a change of domain for the model, from the physical domain (with cartesian base) to the modal domain. **Φ** can be obtained solving the undamped Equation (2), discarding the trivial solution **Φ** = **0**. **Φ** is considered mass-normalized when expression (3) is satisfied.
(2)$$(K-\lambda M)\Phi =(K-\omega^{2}M)\Phi =0$$
(3)$$\Phi^{T}M\Phi =I$$

In its full form, **Φ** contains as many mode shapes as DoFs of the full model, but it is possible to reduce the model removing the modes out of the frequency range of interest (modal truncation). For a *k* number of modes of interest, **Φ** is reduced to **Φ_K_** (4), with its dimension reduced to *n* × *k.*
(4)$$\Phi_{\left(n,k\right)}=[\varphi_{1},\varphi_{2}\ldots \varphi_{k}]$$

Through the Φ-transformation, the dynamic Equation (1) can be transformed into the generalized dynamical Equation (5), where **z**(t) is the vector of modal displacements (also known as generalized displacements), obtained with the transformation **q**(t) = **Φ_K_ z**(t). Equation (5) can also be expressed as (6), **Φ_K_^T^MΦ_K_** being an identity matrix, **Σ** a diagonal matrix containing the damping ratios (ξ) associated with each frequency, and **Ω** the diagonal matrix with the natural frequencies of the model (ω).
(5)$${\Phi_{k}}^{T}M\Phi_{k}\ddot{z}\left(t\right)+{\Phi_{k}}^{T}C_{D}\Phi_{k}\dot{z}\left(t\right)+{\Phi_{k}}^{T}K\Phi_{k}z\left(t\right)={\Phi_{k}}^{T}f(t)$$
(6)$$\ddot{z}\left(t\right)+2\Sigma \Omega \dot{z}\left(t\right)+\Omega^{2}z\left(t\right)={\Phi_{k}}^{T}f(t)$$

#### 2.1.3. State-Space Model

A MCK model can be described as a state-space system (7), that consists of two equations: the state Equation (above) and the output Equation (below). **x** is the state vector, with 2*n* × 1 dimension. As shown in (8), the state vector contains the displacements and the velocities of each DoF. **u** is the input vector, and with *n* × 1 dimension, contains the possible external input for each DoF. **A** and **B** are the state and input matrices, respectively. As it seen in (9) and (10), the dimensions of these matrices are 2*n* × 2*n* and 2*n* × *n*, respectively. The elements of the output equation, the output vector **y** and the output and feedthrough matrices **C** and **D**, change according to the desired output variables. State-space notation is required to implement the model in Kalman filters and variants.
(7)$$\left\{\begin{array}{c} \dot{x}=Ax+Bu\\ y=Cx+Du \end{array}\right.$$
(8)x=qq˙
(9)$$A=\left[\begin{array}{cc} 0 & I\\ {-M}^{-1}K & {-M}^{-1}C_{D} \end{array}\right]$$
(10)$$B=\left[\begin{array}{c} 0\\ M^{-1} \end{array}\right]$$

To use the state-space model in a discrete-time approach, the **A** and **B** matrices must be discretized. **A_d_** (11) and **B_d_** (12) are the discretized versions of the state-space model matrices.
(11)$$A_{d}=e^{A∆t}$$
(12)$$B_{d}=A^{-1}\left(A_{d}-I\right)B$$

### 2.2. Virtual Sensing Algorithms

In this subsection, the selected vs. algorithms in this article are described: the Kalman filter, the Augmented Kalman filter and the least-squares strains estimation. The observability conditions of each algorithm are also described.

#### 2.2.1. Kalman Filter

The KF is a Bayesian recursive algorithm used to estimate the hidden states of a system. A state-space model of the system is used to make predictions of the states, and information coming from a limited number of real sensors is used to correct the predictions.

The KF is an algorithm of stochastic nature that manages gaussian uncertainties associated with the used model and with the measurements. **Q** (13) is the covariance matrix of the model (with 2*n* × 2*n* dimension) and **R** (14) is the covariance matrix of the input sensors (with r × r dimension, being r the number of input sensors). Assuming that the states and the measurements are not correlated with each other, the matrices **Q** and **R** are simplified to diagonal matrices, where each value of the diagonal corresponds to the uncertainty associated with each state (q) and with each sensor input (r), respectively. The **Q** matrix must be discretized when used in a discrete-time Kalman filter (15).
(13)$$Q=diag(q_{1},q_{2}\ldots ,q_{2n})$$
(14)$$R=diag(r_{1},r_{2}\ldots ,r_{r})$$
(15)$$Q_{d}={(A}_{d}Q{A_{d}}^{T})∆t$$

In absence of external force measurements, the KF is implemented as follows: states prediction (16), covariance prediction (17), Kalman gain determination (18), states prediction update (19) and covariance prediction update (20).
(16)$$x_{t}={Ax_{t-1}}_{}$$
(17)$$P_{t}={AP_{t-1}A^{T}+}_{}Q$$
(18)$$K_{t}={P_{t}H^{T}{\left(HP_{t}H^{T}+R\right)}^{-1}}_{}$$
(19)$${x_{t}}^{updated}=x_{t}+K_{t}(z_{t}-Hx_{t})$$
(20)$${P_{t}}^{updated}=P_{t}-K_{t}HP_{t}$$

The incorporation of the real sensor measurements into the filter (described in states prediction update step) is performed with the measurement matrix (**H**). This matrix relates each measurement with their corresponding states. It has r × 2*n* dimension, being 2*n* the number of states of the system and r the number of input sensors.

With a measurement data vector **z**(t) containing *x* number of strain gauges and *y* number of accelerometers, the **H** matrix is built as seen in (21). To relate the strain gauge data to the modal states, the modal strains are obtained from the FE model. These can be obtained from a modal analysis of the FE model of the structure, compiling the strain value (ε) obtained in each gauge (1 to *x*) for each mode (1 to *n*). To relate the accelerometer data to the modal states, the corresponding rows of modal **M**, **C** and **K** matrices are used.

Because of the external force measurements are not available, no relation between the force and acceleration measurements is implemented (in the case that external force measurements were available, these would be related to the acceleration measurements through a **J** matrix (22)). Because of this, the uncertainty of accelerometer measurements is expected to be greater.
(21)$$H=\left[\begin{array}{cc} \begin{array}{ccc} \varepsilon_{1,1} & \cdots & \varepsilon_{1,n}\\ \vdots & \ddots & \vdots \\ \varepsilon_{x,1} & \cdots & \varepsilon_{x,n} \end{array} & \begin{array}{ccc} 0 & \cdots & 0\\ \vdots & \ddots & \vdots \\ 0 & \cdots & 0 \end{array}\\ \begin{array}{ccc} {{-M}_{1,1}}^{-1}K_{1,1} & \cdots & {{-M}_{1,n}}^{-1}K_{1,n}\\ \vdots & \ddots & \vdots \\ {{-M}_{y,1}}^{-1}K_{y,1} & \cdots & {{-M}_{y,n}}^{-1}K_{y,n} \end{array} & \begin{array}{ccc} {{-M}_{1,1}}^{-1}C_{1,1} & \cdots & {{-M}_{1,n}}^{-1}C_{1,n}\\ \vdots & \ddots & \vdots \\ {{-M}_{y,1}}^{-1}C_{y,1} & \cdots & {{-M}_{y,n}}^{-1}C_{y,n} \end{array} \end{array}\right]$$
(22)$$J=\left[\begin{array}{c} \begin{array}{ccc} 0 & \cdots & 0\\ \vdots & \ddots & \vdots \\ 0 & \cdots & 0 \end{array}\\ \begin{array}{ccc} {M_{1,1}}^{-1} & \cdots & {M_{1,n}}^{-1}\\ \vdots & \ddots & \vdots \\ {M_{y,1}}^{-1} & \cdots & {M_{y,n}}^{-1} \end{array} \end{array}\right]$$

In a KF, observability can be defined as the capacity of the algorithm to obtain enough information from the real system (through the input sensors and the observation matrix) to be able to estimate all the states. To determine if a KF is observable, the observability matrix **O** (23) is calculated using the transpose of **A**. Only if the rank of **O** is equal to 2*n* (the number of states of the model) is the KF is fully observable.
(23)$$O=\left[\begin{array}{c} \begin{array}{c} A^{T}H^{0}\\ A^{T}H^{1} \end{array}\\ \begin{array}{c} \vdots \\ A^{T}H^{2n-1} \end{array} \end{array}\right]$$

#### 2.2.2. Augmented Kalman Filter

The AKF is a variant of the KF in which the external forces over the system are considered additional states of the model. Thanks to this feature, this filter does not need the external force applied on the monitored system as input. The AKF uses an augmented state-space model of the system that combines the **A** and **B** matrices of the state-space model in a single matrix **A*** (24) with (2*n* + *n_f_*) × (2*n* + *n_f_*) dimension (*n_f_* being the number of expected external forces), and an augmented vector of states **x*** (25) that combines the displacements, their first derivatives and the external input forces (resulting in a 2*n* + *n_f_* dimension). The discretization of **A*** is shown in (26).
(24)$$A^{*}=\left[\begin{array}{cc} A & B\\ 0 & 0 \end{array}\right]$$
(25)$$x^{*}=\left(\begin{array}{c} q\\ \dot{q}\\ u \end{array}\right)$$
(26)$${A_{d}}^{*}=\left[\begin{array}{cc} A_{d} & B_{d}\\ 0 & I \end{array}\right]$$

The unknown input is modeled as a zero-mean random walk model, so the covariance matrix of the model **Q** must be augmented to (2*n* + *n_f_*) × (2*n* + *n_f_*) dimension by adding a term related to the uncertainty associated to the external forces (27).
(27)$$Q^{*}=\left[\begin{array}{cc} Q_{d} & 0\\ 0 & Q_{u} \end{array}\right]$$

An augmented observation matrix **H*** (28) must be defined by combining the observation matrix **H** (21) and the input observation matrix **J** (22), resulting in a matrix of *r* × (2*n* + *n_f_*) dimension.
(28)$$H^{*}=\left[\begin{array}{cc} H & J \end{array}\right]$$

In the AKF, observability has the same meaning as in the classical KF. To determine if an AKF is observable, the observability matrix **O*** (29) must be calculated. Only if the rank of **O*** is equal to 2*n* + *n_f_* is the AKF is fully observable.
(29)$$O^{*}=\left[\begin{array}{c} \begin{array}{c} A^{*T}H^{*0}\\ A^{*T}H^{*1} \end{array}\\ \begin{array}{c} \vdots \\ A^{*T}H^{*2n-1} \end{array} \end{array}\right]$$

#### 2.2.3. Least-Squares Strain Estimation (LSSE)

The LSSE is a deterministic virtual sensing algorithm that uses a matrix generalized inversion to obtain the least squares solution of the unknown strains. The Moore-Penrose pseudoinverse [20] and the Modal Expansion [24] are used for this purpose. This method allows obtaining strain estimates at unmeasured points both in the presence and absence of dynamic effects.

The linear equation is stated by relating the measured strain and the modal displacements of the system (30). In a linear system, displacements **x**(t) and measured strains **z_i_**(t) are linearly related through the modal strain matrix **G_i_** (with *g* × *m* dimension, *g* being the number of strain measurements, and m the number of modal displacements).
(30)$$z_{i}\left(t\right)=G_{i}x(t)$$

Using the same statement, strain virtual measurements **z_vs_**(t) can be obtained from the modal displacements, through the modal strain matrix **G_vs_** (with *o* × *m* dimension, *o* being the number of virtual strain sensors, and *m* the number of modal displacements) (31).
(31)$$z_{vs}\left(t\right)=G_{vs}x(t)$$

Using the pseudoinverse of **G_i_**, both statements can be combined to obtain strain virtual measurements from a set of real strain measurements (32).
(32)$$z_{vs}\left(t\right)=G_{vs}\left[{G_{i}}^{+}z\left(t\right)\right]$$

If the number of strain measurements *g* is equal to the number of modal displacements *m*, the statement (31) is determined, and the solution is found by the LSSE. If *g* is higher than *m*, the statement is overdetermined. If, on the contrary, *g* is lower than *m*, the statement (31) is underdetermined. In both cases, the LSSE gives a best-fit approximation of the solution. To provide a good approximation of the solution, the condition number of the matrix **G_i_** must be close to 1. If the condition number of **G_i_** is high, the statement (31) is ill conditioned and significant errors can be expected in the solution.

### 2.3. Virtual Sensing Implementation

The selected vs. algorithms are tested on a use case defined in Section 2.4. First, an FE model of the use case is created. This model is used to choose the location of the sensors (strain gauges and accelerometers) in the real prototype. Measurement data obtained from the sensors is first used to adjust and validate the model, and then to feed the vs. algorithms. The obtained estimations are compared to the equivalent measurement data to evaluate the performance of the vs. algorithms under the different conditions. The entire process is summarized in the flowchart shown in Figure 1.

### 2.4. Use Case

In this subsection, the use case, the installed sensors, and the modelling process are described.

#### 2.4.1. Prototype Description

The use case is a scaled wind turbine tower prototype installed on a jacket-type structure, which is fixed to the ground (Figure 2). The main specifications of the prototype can be seen in Table 1. An electromagnetic inertial shaker, considered as part of the system, is placed on top of the prototype attached to a rotating platform to excite the structure in different directions (Figure 3) and frequency components (<25 Hz). The specifications of the shaker used can also be seen in Table 1.

#### 2.4.2. FE Model and Model Reduction

A FE model of the prototype is built based on the 3D CAD of the structure. The shaker and its support, including the bushing, are simplified to an equivalent point mass located at the mass center of the replaced components and attached to the structure. The behavior of the bushing has been tested in the frequency range of interest (0 to 25Hz), verifying its linearity. The platform has been designed to keep the mass center of the rotating components in the rotating axis, so the system can be considered invariant. The bolted joints present in the prototype are also simplified using bonded contacts. Due to the tower and nacelle of the wind turbine prototype being thin steel profile components, shell-type elements have been used to reduce the total number of elements in the mesh. In the jacket support structure, solid elements have been used. The primary features of the FE model can be seen in Table 2, and the FE model of the prototype can be seen in Figure 4.

Modal truncation is applied to obtain a reduced model. Following the criteria indicated in the Section 2.2, the minimum number of modes to include are the first 4 modes, because their frequencies are inside the range of the external forces applied (between 2 and 25 Hz) and accumulate more than 90% of the modal mass. The 5th and following modes have frequencies over 50 Hz.

#### 2.4.3. Model Validation

To compare the modal response of the model to the real prototype, an operational modal analysis (OMA) has been performed using two accelerometers located in the nacelle of the prototype (with the shaker off). The current data acquisition system has been used, in combination with the software PULSE. All the first four modes of the model are detected in the real prototype. The stiffness of the contacts between tower segments in the model has been adjusted in order to obtain a better modal adjustment.

The four detected modes have good MAC correlation values with the model modes (all above 85%). The relative error between the frequencies of the model and the frequencies of the detected prototype model are below 10% in all cases. The correlations between the modes of the FE model and the detected modes in the real prototype are shown in Table 3. The four modal shapes are shown in Figure 5.

#### 2.4.4. Sensors

The set of sensors installed in the prototype consists of strain gauges, accelerometer, and a potentiometer to measure the shaker direction. No sensors for force measurement have been installed in the prototype. The location of the gauges and the accelerometers has been chosen following the Modal Kinetic Energy method [33]. A sufficient number of gauges have been installed to be able to work with different configurations of input sensors, keeping reference sensors to have reference real data in the locations of the virtual sensors. All the used strain gauges measure the normal strain in the axial (Z) direction. A triaxial accelerometer has been located at the top of the prototype, the zone with maximum displacement. The number and main specifications of the sensors used can be found in Table 4, and their location can be seen in Figure 4.

The data acquisition system used is the software LabVIEW of National Instruments (NI) in combination with the hardware NI CompactDAQ-9189, a chassis with 8 slots for acquisition modules. Modules NI-9235 have been used for the quarter bridge gauges, NI-9237 for the half bridge gauges, NI-9234 for the accelerometers and NI-9201 for the angle sensor. 

Prior to carrying out the experiments, the sensors were calibrated. The accelerometer has been calibrated both statically and dynamically, checking its response to orientation changes for the first case, and using a calibration shaker (PCB 394C06 handheld shaker) for the second case. The strain gauges have been calibrated by first checking the offset value (with the prototype unloaded), then applying static measured forces to the prototype (using a dynamometer) and checking their response with the FE model strain response at the same points. The uncertainty value have been measured in 0.3 µm/m in the case of the strain gauges, and in 0.01 m/s^2^ in the case of the accelerometer.

## 3. Results

In this section, the estimations obtained with the vs. algorithms are shown. The evaluation methods (for comparing the estimated data with the measured data of reference), as well as all the input forces and input sensors configurations used, are also described.

### 3.1. Evaluation Methods

The estimates obtained in the virtual sensors are evaluated using a set of indicators, first in order to determine if the virtual sensors are within an acceptable reliability range, and secondly to be able to compare the performance of the different virtual sensing algorithms tested using different configurations of input sensors.

The mean of the error signal (the subtract from the estimated signal to the reference signal) indicates a bias in the estimated signal, which should be close to 0.

A certain delay between the estimated signals and the reference is expected, especially when KF and AKF are used. In this study, the delay in the estimations is not considered relevant, so when some correlation parameters are used (those where the presence of a delay would alter the results) the existing delay is calculated and compensated. The Pearson correlation coefficient (PCC) is a normalized measurement of the covariance between two signals (33). This indicator is obtained through the quotient of the cross covariance of the estimated signal (cov (est, ref)) and the reference signal with the product of the standard deviations of the reference signal and the estimated signal (σ_est_ σ_ref_). The PCC has a rank between 100% and −100%: a PCC = 100% means that the estimation of the virtual sensor is fully correlated with the real measurement, a PCC = 0% means that there is no correlation between the estimate and the real measurement, and a PCC = −100% means that there is an inverse correlation. The PCC is used to evaluate if the estimated signals (with delay compensated) are consistent in shape with the reference signals.
(33)$${PCC}_{est,real}=\frac{cov(est,ref)}{\sigma_{est}\sigma_{ref}}\times 100$$

The percentage error of the estimations is obtained using the formula (34).
(34)$$error\left[\%\right]=\left|1-\frac{\sigma_{est}}{\sigma_{ref}}\right|\times 100$$

For the purpose of this article, strain estimates are considered acceptable when they have a percentage error below 10% in the cases of forces applied in X or Y directions, and below 20% in the cases of forces applied in combined directions. A PCC above 90% is considered acceptable in all cases.

### 3.2. Tests Performed

To test the virtual sensing algorithms, an organized sequence of tests has been used. The different algorithms with different input sensor configurations are tested using different external input forces applied in different directions. The external input forces used in the tests are collected in Table 5. A few examples of results with different external inputs (such as variable direction or hammer impacts) are shown in Figures 9 and 10.

With the aim of summarizing the considerable number of obtained results, two reference sensors are used, in which the performance of the algorithms is studied: the gauges 1-X-90 and 1-Y-90, at the bottom of the tower. These gauges have been selected as reference gauges because is where a greater response is expected. The rest of the sensors are used as input sensors for the algorithms. All tested configurations are collected in Table 6.

For each sensor configuration tested, KF parameters (R, Q and QF values) are adjusted through an iterative process. The R values of the sensor measurements and the QF value of the augmented states of the AKF are held constants for all the configurations used, while the Q values are varied for each sensor configuration in order to achieve the highest accuracy in the virtual sensor estimations. The selected R values can be seen in Table 7, and the selected Q and QF values can be seen in the Table 8. The input sensor data used is previously filtered using a 26 Hz low-pass Butterworth filter.

### 3.3. Obtained Results

For each configuration and for each direction of application of the input forces, the results shown are an average of the estimates obtained for each input force applied. A limit value of 4 µm/m in the standard deviation of the reference signals has been set, based on the experience obtained from the experimental tests. Above that limit, the estimation of that reference signal is considered in the average. Below that limit, the reference signal is considered background noise and its estimation is not taken into account in the average.

The results obtained with the KF, AKF and LSSE are summarized in Table 9, Table 10 and Table 11, respectively. 

## 4. Discussion

In total, 12 different sensor configurations have been tested in the KF and AKF algorithms: five configurations with only strain gauges, and seven configurations combining strain gauges and an accelerometer. In the case of the LSSE, since only admits strain gauge signals as inputs, only the first five configurations have been tested. No force measurements are available. A modal-truncated model with four modes has been used for all cases.

Firstly, it has been observed that when only strain gauges are used as input sensors, the three tested algorithms (KF, AKF and LSSE) perform remarkably similarly. Only in certain cases (such as the estimations obtained under variable direction forces), the KF shows some deviation from the other two algorithms. When an accelerometer is added to the configuration, the KF fails, and a large level of error is detected in the estimations. In the case of the AKF, the addition of an accelerometer as sensor input does not bring a significant improvement. In some cases, the error detected in the virtual sensors increases appreciably with the included accelerometer. This can be explained because, in the absence of force measurements, the uncertainty associated with the accelerometer measurements grows significantly, as was predicted in Section 2.2.1.

The error detected in the virtual sensors does not always increase as the number of sensors of the configuration is reduced. For example, when an eight-strain-gauge configuration is used (the maximum number available), greater error is detected than when a six-strain-gauge configuration is used. When the number of gauges is reduced to four or less, the detected error grows significantly. This behavior coincides in the three tested algorithms. It is also observed that not only the number of input sensors in the configuration, but also their position, influence the response of the virtual sensing algorithms. This agrees with the fact that some sensor locations are more adequate for measuring the system behavior, as is predicted by sensor location methods (such as the Modal Kinetic Energy method, mentioned in Section 2.4.4).

According to the obtained results in the virtual sensors when forces in X direction are applied, the estimates obtained in the virtual sensors are generally better than when forces in the Y direction or in combined directions are applied. This difference can be explained by the certain lack of precision in the behavior of the FE model (from which the reduced model has been obtained) in the Y-direction bending and the Z-direction torsion. Of the sensor configurations tested in the different vs. algorithms, the best-performing one is configuration 3. This configuration uses the 3X, 3Y, 4X, 4Y, 5X and 5Y gauges as input sensors. Configuration 8, that uses the same input gauges but adding an accelerometer, also performs well when is used with the AKF.

From the data obtained from the experiments carried out in this article, it can be concluded that, in terms of robustness, the LSSE is preferable because, unlike Kalman filters, it does not depend on tuning parameters. Among the Kalman filters, the AKF can be considered more robust than the KF because, under conditions of unmeasured forces, it has a stable performance when accelerometers are used as input sensors.

Some examples of the obtained results applying different input forces, using configuration 3 (input gauges X-2-90, Y-2-90, X-3-90, Y-3-90, X-4-90, Y-4-90, X-5-90, Y-5-90), are provided in Figure 6, Figure 7, Figure 8, Figure 9 and Figure 10. The values corresponding to these results are summarized in Table 12, Table 13, Table 14, Table 15 and Table 16. The virtual sensors are the gauges 1X (first column of the tables) and 1Y (second column of the tables). The estimations obtained with the KF, AKF and LSSE are compared with real strain data at the same location (indicated as REF). The left values in the tables correspond to the percentage error of the estimation, and the right values correspond to the PCC error.

**Figure 6 sensors-23-04706-f006:**
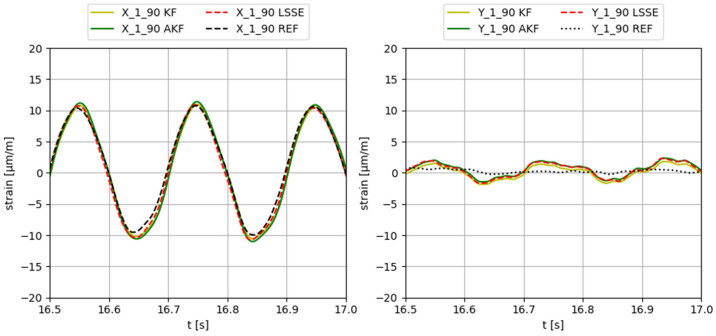
Results obtained applying a 5Hz sinusoidal force in X direction.

**Figure 7 sensors-23-04706-f007:**
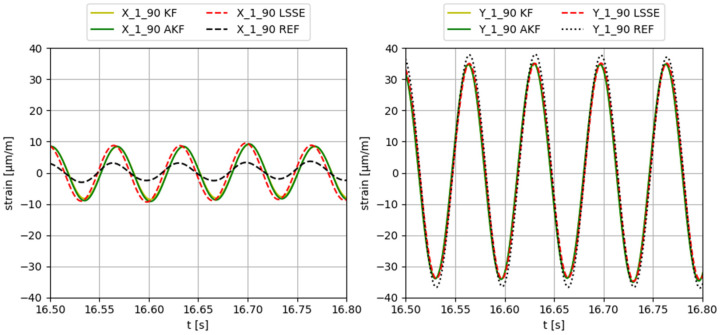
Results obtained applying a 15 Hz sinusoidal force in Y direction.

**Figure 8 sensors-23-04706-f008:**
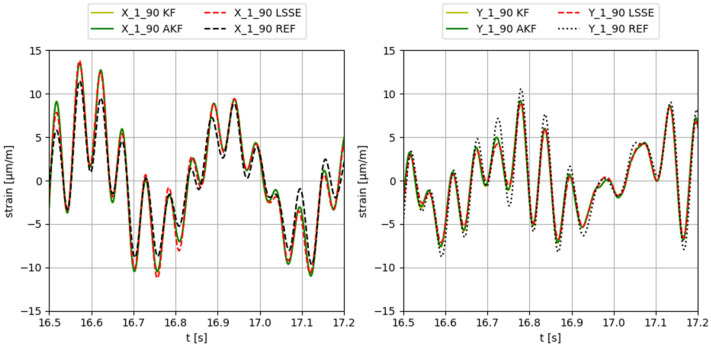
Results obtained applying a zero-mean white noise force in 45° direction.

**Figure 9 sensors-23-04706-f009:**
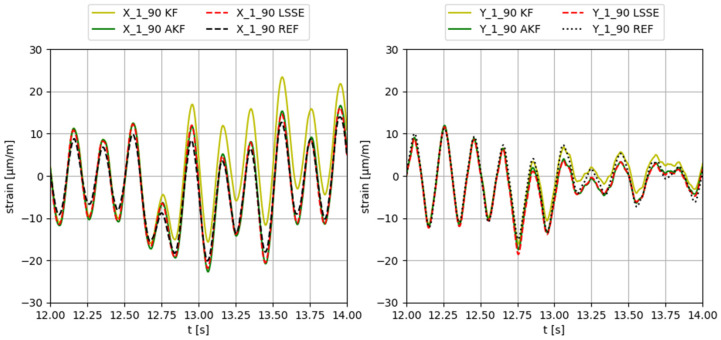
Results obtained applying a 5 Hz sinus force in variable direction along time.

**Figure 10 sensors-23-04706-f010:**
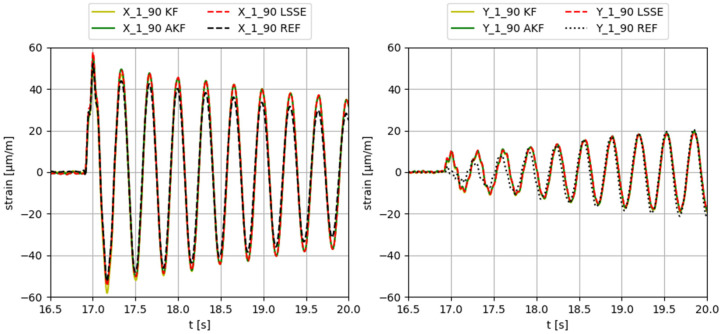
Results obtained applying a hammer impact in X direction.

## 5. Conclusions

In this article, three different vs. algorithms have been tested to obtain virtual strain estimations under unknown forces. Twelve different sensor configurations have been used under 15 different dynamic loads. Two virtual strain gauges have been implemented in the base of the tower of the prototype, and two real strain gauges have been used as reference sensors, to compare the estimated data obtained from the virtual sensors with the equivalent real sensor data.

It has been verified that, through the modal truncation, a reduced model can be used to obtain the response of a much more complex FE model (in a limited range of frequencies) using a limited number of modes (which implies a significative reduction in the number of DoFs used). The AKF shows itself to be better than the classical KF in absence external force measurements, especially when strain and acceleration measurements are available. If only strain measurements are available, the AKF and the LSSE perform similarly, so, to obtain strain virtual measurements, the LSSE may be preferable due to its simplicity.

The experience and results obtained with the experiments presented in this article can be useful when implementing strain virtual sensors. Examples of application can range from wind turbines (as in the case of this article) to many other types of complex structural assets, for example distinct types of offshore structures, bridges, communication towers or even large industrial frames (such as industrial presses).

Several future lines of inquiry may continue the work presented in this article. On one hand, it would be interesting to add real-time force measurements to allow comparison vs. results with known forces and with unknown forces. Forces can also be estimated using vs. algorithms. On the other hand, it would also be interesting to install more accelerometers to the use case, in order to be able to test more sensor configurations in the different vs. algorithms. It would be also interesting to install gauges with an orientation of 45º with respect to the tower axis, with the aim of measuring torsional strain. Furthermore, it would be interesting to apply the tested vs. algorithms in other use cases of a different nature, to check if vs. estimations of comparable quality can be obtained in other types of structures.

## Figures and Tables

**Figure 1 sensors-23-04706-f001:**
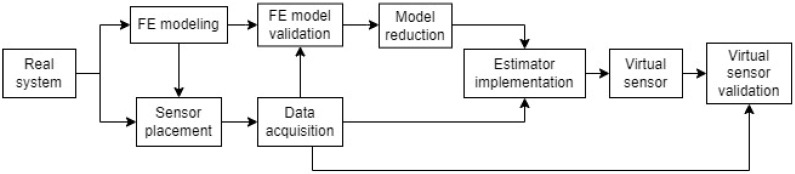
Flowchart of the process followed to implement and test vs. algorithms.

**Figure 2 sensors-23-04706-f002:**
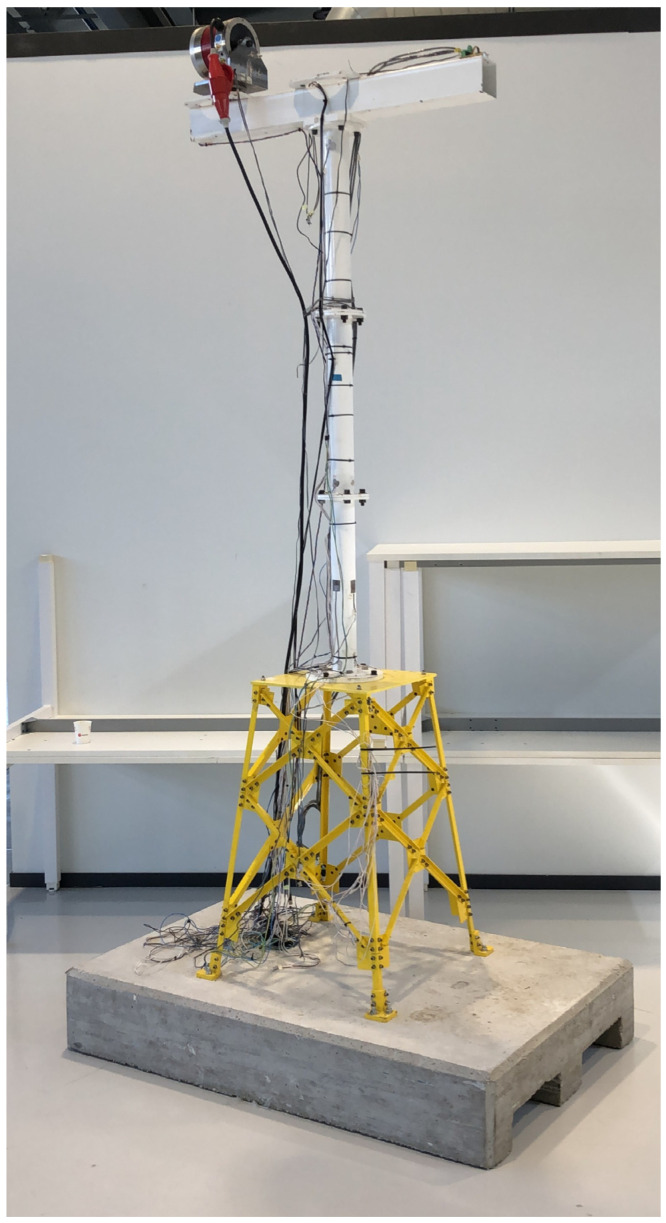
General view of the prototype and the concrete foundation.

**Figure 3 sensors-23-04706-f003:**
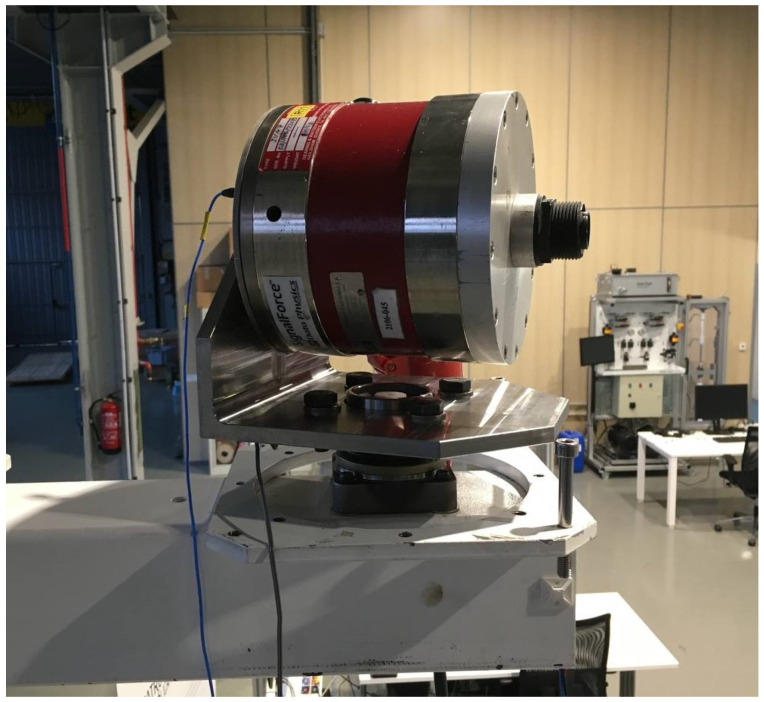
Inertial shaker Data Physics IV47 attached on top of the prototype.

**Figure 4 sensors-23-04706-f004:**
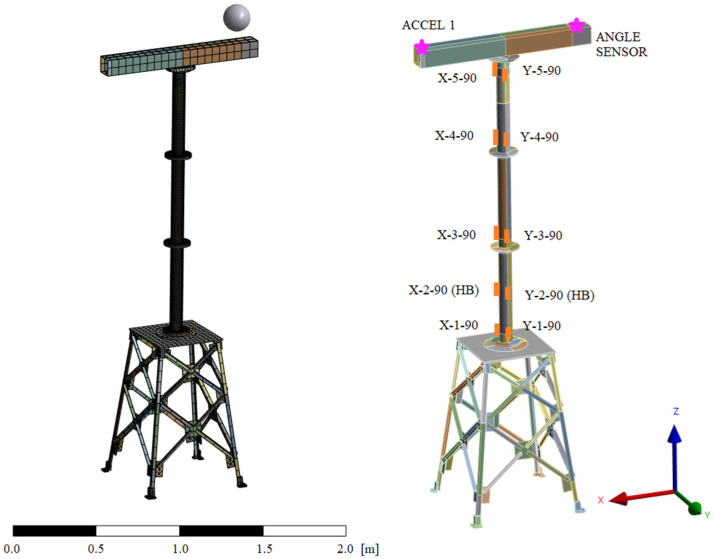
FE model where the shaker and its support are simplified in a point mass (**left**) and sensor locations on the prototype (**right**).

**Figure 5 sensors-23-04706-f005:**
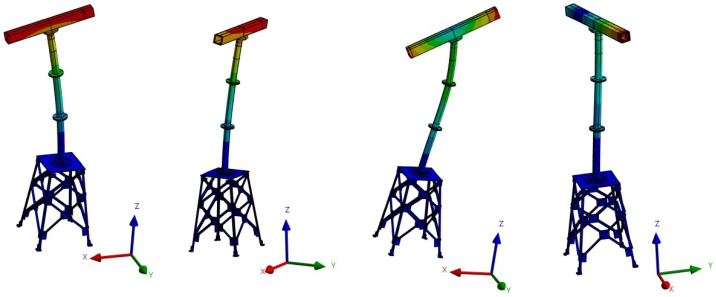
Modal shapes of the first 4 modes.

**Table 1 sensors-23-04706-t001:** Main specs of the prototype and the shaker.

Feature	Value
Tower + nacelle weight	42 kg
Jacket weight	13.5 kg
Shaker + support weight	27 kg
Total weight	82.5 kg
Tower height	1790 mm
Jacket height	1300 mm
Total height	3090 mm
Material	Steel
Supports	Fixed to a concrete base
Shaker model	Data Physics IV47
Inertial mass	14.5 kg
Max sinus force (peak)	250 N
Total shaker mass	21 kg
Shaker main mode freq.	20 Hz

**Table 2 sensors-23-04706-t002:** Main specs of the FE model.

Feature	Description
Number of elements	11,387
Element type	Shell and solid
Element order	Quadratic
Young modulus	2.2 × 10^11^ Pa
Boundary conditions	4 fixed supports
Model weight	82.5 kg

**Table 3 sensors-23-04706-t003:** Correlation between the modes of the model and modes detected in the real prototype.

FE Model			OMA		
Mode	Mode Description	freq. [Hz]	freq. [Hz]	Damping Ratio [%]	MAC [%]
1	1st in X. Bending	3.1	3.1	0.5	85
2	1st in Y. Bending	3.1	3.1	0.7	93
3	2nd in X. Bending	17.0	16.2	4.1	86
4	1st in Z. Torsion	19.2	17.4	2.7	99

**Table 4 sensors-23-04706-t004:** Number and specifications of installed sensors.

Sensor	Number	Description
Strain gauges: quarter bridge	8	120 Ω, gauge factor = 2
Strain gauges: half bridge	2	120 Ω, gauge factor = 2 *
Accelerometer	1 (3 channels)	MEMS type (ADXL335)
Angle sensor	1	Potentiometer

* Each individual gauge.

**Table 5 sensors-23-04706-t005:** External inputs used in the performed tests.

Input Force Type	Frequency [Hz]	Directions
0 mean sinusoidal	2	X, Y and XY
0 mean sinusoidal	5	X, Y and XY
0 mean sinusoidal	10	X, Y and XY
0 mean sinusoidal	15	X, Y and XY
0 mean white noise	5 to 25	X, Y and XY

**Table 6 sensors-23-04706-t006:** Input sensor configurations tested. The “G” column indicate the use of the strain gauges (both X and Y) corresponding to the height of the tower shown in Figure 5. The “Ac”columns indicate the use of the channels of the accelerometer 1.

Algorithms	Config.	AcX	AcY	Ac Z	G2	G3	G4	G5
KF, AKF, LSSE	1				x	x	x	x
KF, AKF, LSSE	2				x	x	x	
KF, AKF, LSSE	3					x	x	x
KF, AKF, LSSE	4				x	x		
KF, AKF, LSSE	5						x	x
KF, AKF	6	x	x	x	x	x	x	x
KF, AKF	7	x	x	x	x	x	x	
KF, AKF	8	x	x	x		x	x	x
KF, AKF	9	x	x	x	x	x		
KF, AKF	10	x	x	x			x	x
KF, AKF	11	x	x	x	x			
KF, AKF	12	x	x	x				x

**Table 7 sensors-23-04706-t007:** R values for each sensor used.

R	Value
gauges	1 × 10^8^
accelerometers	1 × 10^3^

**Table 8 sensors-23-04706-t008:** Q and QF tuning parameters for each configuration. The number of truncated modes used is also included.

Configuration	KF		AKF			LSSE
	Modes	Q	Modes	Q	QF	Modes
1	4	1 × 10^−1^	4	1 × 10^−1^	1 × 10^3^	4
2	4	1 × 10^−1^	4	1 × 10^−1^	1 × 10^3^	4
3	4	1 × 10^−1^	4	1 × 10^−1^	1 × 10^3^	4
4	4	1 × 10^−1^	4	1 × 10^−1^	1 × 10^3^	4
5	4	1 × 10^−2^	4	1 × 10^−2^	1 × 10^3^	4
6	4	1 × 10^−9^	4	1 × 10^−1^	1 × 10^3^	-
7	4	1 × 10^−9^	4	1 × 10^−1^	1 × 10^3^	-
8	4	1 × 10^−9^	4	1 × 10^−1^	1 × 10^3^	-
9	4	1 × 10^−9^	4	1 × 10^−1^	1 × 10^3^	-
10	4	1 × 10^−9^	4	1 × 10^−1^	1 × 10^3^	-
11	4	1 × 10^−9^	4	1 × 10^−1^	1 × 10^3^	-
12	4	1 × 10^−9^	4	1 × 10^−3^	1 × 10^3^	-

**Table 9 sensors-23-04706-t009:** KF results.

Configuration	Number Gauges	Number Accelerometers	Error %/PCC X Direction	Error %/PCC Y Direction	Error %/PCC XY 45° Direction
1	8	0	11.0/99.3	19.2/99.9	16.9/99.5
2	6	0	10.4/99.2	19.4/99.8	19.3/99.3
3	6	0	8.8/97.8	5.6/99.7	19.4/99.4
4	4	0	13.4/99.3	23.7/99.8	21.8/99.1
5	4	0	30.3/84.6	22.0/96.4	28.1/97.2
6	8	1	751.5/86.4	55.9/73.5	368.8/81.2
7	6	1	751.5/86.4	55.9/73.5	368.8/81.2
8	6	1	751.5/86.4	55.9/73.5	368.8/81.2
9	4	1	751.5/86.4	55.9/73.5	368.8/81.2
10	4	1	751.5/86.4	55.9/73.5	368.8/81.2
11	2	1	54.3/91.5	56.3/73.2	50.8/90.4
12	2	1	54.3/91.5	56.3/73.2	50.8/90.4

**Table 10 sensors-23-04706-t010:** AKF results.

Configuration	Number Gauges	Number Accelerometers	Error %/PCC X Direction	Error %/PCC Y Direction	Error %/PCC XY 45° Direction
1	8	0	10.6/99.4	19.2/99.9	17.3/99.7
2	6	0	9.9/99.2	19.4/99.8	19.6/99.3
3	6	0	7.3/97.9	4.6/99.8	19.2/98.4
4	4	0	13.3/99.3	24.2/99.8	22.0/99.1
5	4	0	32.4/87.0	16.6/97.9	26.7/97.1
6	8	1	10.3/99.0	26.3/99.5	22.6/98.7
7	6	1	15.1/99.0	24.2/99.6	21.8/98.5
8	6	1	4.5/97.1	18.4/98.4	19.3/97.5
9	4	1	13.6/99.4	63.0/74.3	35.4/94.3
10	4	1	35.1/81.3	65.2/65.2	47.1/76.6
11	2	1	56.6/82.7	66.6/77.7	53.9/91.2
12	2	1	70.2/46.3	87.1/49.3	76.6/70.3

**Table 11 sensors-23-04706-t011:** LSSE results.

Configuration	Number Gauges	Error %/PCC X Direction	Error %/PCC Y Direction	Error %/PCC XY 45° Direction
1	8	11.1/99.5	20.1/99.9	17.5/99.7
2	6	10.9/99.3	21.5/99.9	20.9/99.4
3	6	8.2/99.6	5.5/99.8	18.8/98.9
4	4	15.7/99.2	30.5/99.7	25.7/98.9
5	4	36.4/84.1	26.0/97.3	45.2/90.0

**Table 12 sensors-23-04706-t012:** Evaluation of the results shown in Figure 6.

F X 5 Hz	X-1-90	Y-1-90
KF	3.6/99.4	178.6/61.6
AKF	7.7/99.4	187.3/60.2
LSSE	4.1/99.8	197.1/58.3

**Table 13 sensors-23-04706-t013:** Evaluation of the results shown in Figure 7.

F Y 15 Hz	X-1-90	Y-1-90
KF	195.6/95.4	7.3/99.9
AKF	209.3/95.2	7.0/99.9
LSSE	217.3/98.5	6.9/99.9

**Table 14 sensors-23-04706-t014:** Evaluation of the results shown in Figure 8.

F XY Noise	X-1-90	Y-1-90
KF	19.8/97.4	13.7/99.1
AKF	20.0/97.4	13.9/99.0
LSSE	18.2/97.3	15.9/98.9

**Table 15 sensors-23-04706-t015:** Evaluation of the results shown in Figure 9.

F VAR 5 Hz	X-1-90	Y-1-90
KF	12.3/92.8	8.6/99.0
AKF	19.5/99.0	4.9/98.9
LSSE	16.8/99.3	4.1/98.7

**Table 16 sensors-23-04706-t016:** Evaluation of the results shown in Figure 10.

F X HIT	X-1-90	Y-1-90
KF	11.6/99.5	2.1/93.7
AKF	11.3/99.5	1.8/93.3
LSSE	11.6/99.6	3.2/92.4

## Data Availability

The data presented in this study are available on request from the corresponding author. The data are not publicly available due to privacy restrictions.

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
