# Peer review of "Strain Virtual Sensing for Structural Health Monitoring under Variable Loads"

_sensors, 2023, doi:10.3390/s23104706_

Round 1
Reviewer 1 Report
Major comments
1 The reviewer believe that the introduction of VS algorithms are not adequate. In section 1, the authors only introduce the algorithms which are utlized in this manuscript. Other important algorithms, for example, Bayesian estiamtion or some AI based method are missed.
2. There are some many modified KF algorithms. Why the author choose to use AKF in this manuscript? It is only because that AKF do not need as the external force as input?
3. Is the model reduction necessary for every FE model? How many DoFs are suitbale for the algorithms in this manuscript?
4. In the FE model, why use shell element? Please explain it.
5. In resuslt section, could the author give some disscussion about the robustness of these three algorithms?
Minor comments
1. In line 109, a dot is missed and M, C, D K should be Bold.
There are still some small problem in English expression and spelling. Please check it.
Reviewer 2 Report
The authors presented a study on the use of Kalman filters and Least Squares Estimation algorithms for indirect measurement of strains based on available measured strain and acceleration data from a scaled wind turbine structure in the lab. Results presented in the study suggest that it is possible to perform such indirect measurements if a well calibrated FE model of the structure is available. It turned out that inclusion of acceleration measurements leads to a very large increase of the errors in the estimates.
The reviewer believes that the paper is well written and can be considered for publication after authors will address the following queries.
1. Please provide more details about acceleration measurements. You used very cheap MEMS accelerometers that do not have individual calibration certificates from the manufacturer.
a. How were these sensors calibrated before the experiment?
b. Please provide details of the data acquisition system used for performing operational modal analysis.
2. Please provide estimates of design stage measurement uncertainties for both strain and acceleration in your experimental setup.
3. Please include descriptions of all lines (legends) in Figures 7-10.
4. Please improve resolution and graphics quality in plots in Figures 6-10. It is best to use PNG image format and a resolution of 300 dpi for Word/PPT documents.
Round 2
Reviewer 1 Report
The authors have well revised the manuscript according to my comments.
Reviewer 2 Report
Thank you authors for addressing the comments of the reviewer.
Lines 373-374: " both static in dynamic" - please check the sentence, these words should probably be adverbs: " statically and dynamically".